# Weakly Supervised Semantic Segmentation in Aerial Imagery via Cross-Image Semantic Mining

**Ruixue Zhou** [1,2,3], **Zhiqiang Yuan** [1,2,3], **Xuee Rong** [1,2,3], **Weicong Ma** [4], **Xian Sun** [1,3], **Kun Fu** [1,3] and **Wenkai Zhang** [1,3,*]

1   Aerospace Information Research Institute, Chinese Academy of Sciences, Beijing 100190, China
2   School of Electronic, Electrical and Communication Engineering, University of Chinese Academy of Sciences, Beijing 100190, China
3   Key Laboratory of Network Information System Technology (NIST), Aerospace Information Research Institute, Chinese Academy of Sciences, Beijing 100190, China
4   Xi'an Institute of Applied Optics, Xi'an 710065, China
*   Correspondence: zhangwk@aircas.ac.cn; Tel.: +86-10-5888-7208

**Abstract:** Weakly Supervised Semantic Segmentation (WSSS) with only image-level labels reduces the annotation burden and has been rapidly developed in recent years. However, current mainstream methods only employ a single image's information to localize the target and do not account for the relationships across images. When faced with Remote Sensing (RS) images, limited to complex backgrounds and multiple categories, it is challenging to locate and differentiate between the categories of targets. As opposed to previous methods that mostly focused on single-image information, we propose CISM, a novel cross-image semantic mining WSSS framework. CISM explores cross-image semantics in multi-category RS scenes for the first time with two novel loss functions: the Common Semantic Mining (CSM) loss and the Non-common Semantic Contrastive (NSC) loss. In particular, prototype vectors and the Prototype Interactive Enhancement (PIE) module were employed to capture semantic similarity and differences across images. To overcome category confusions and closely related background interferences, we integrated the Single-Label Secondary Classification (SLSC) task and the corresponding single-label loss into our framework. Furthermore, a Multi-Category Sample Generation (MCSG) strategy was devised to balance the distribution of samples among various categories and drastically increase the diversity of images. The above designs facilitated the generation of more accurate and higher-granularity Class Activation Maps (CAMs) for each category of targets. Our approach is superior to the RS dataset based on extensive experiments and is the first WSSS framework to explore cross-image semantics in multi-category RS scenes and obtain cutting-edge state-of-the-art results on the iSAID dataset by only using image-level labels. Experiments on the PASCAL VOC2012 dataset also demonstrated the effectiveness and competitiveness of the algorithm, which pushes the mean Intersection-Over-Union (mIoU) to 67.3% and 68.5% on the validation and test sets of PASCAL VOC2012, respectively.

**Keywords:** weakly supervised semantic segmentation; remote sensing images; image-level labels

## 1. Introduction

As a result of recent advances in deep convolutional neural networks with powerful feature learning capabilities, semantic segmentation has achieved significant success, including ASPP [1], PSPNet [2], DeepLabV3 [3], etc. However, their application scenes are still constrained by the large amount of pixel-level manual annotations. There has been increasing interest in Weakly Supervised Semantic Segmentation (WSSS) methods, which employ scribble annotations [4,5], bounding boxes [6–9], or image-level annotations [10–13] instead of pixel-level annotations for the purpose of reducing the annotation burden. Compared to other annotations, image-level tags are the least expensive to acquire because they only provide classification cues without any object shape or texture information. This paper

discusses WSSS methods that are based on the most-challenging annotations at the image level in Remote Sensing (RS) scenes.

Most current WSSS models that only utilize image-level labels employ Class Activation Maps (CAMs) [14] to provide initial target localization and generate pseudo-masks, then train the fully supervised semantic segmentation model with the pseudo-masks. However, the trained classifier prefers to focus on the most-discriminative components as opposed to the full objects, which is insufficient for the pseudo-masks required for subsequent semantic segmentation tasks. To address this problem, researchers have made many breakthroughs [15–19] in Natural Scenes (NSs) to boost the segmentation performance of image-level WSSS models. These algorithms, however, only use a single image to localize the target, ignoring the extensive semantic-related context between weakly annotated data, and the image processing is independent of each another. When confronted with RS image scenes containing multiple categories, multi-category localization is not only used to find the targets, but also to distinguish between different categories; however, existing methods perform poorly on multi-category RS datasets.

Therefore, we propose CISM, a novel RS cross-image semantic mining WSSS framework, which can resolve some ambiguities and prevent false predictions by connecting correlated regions across images. Furthermore, propagating correlated representations across images can assist the network in learning more consistent representation capabilities from the whole dataset to obtain more complete target activation regions. The proposed method was demonstrated to be effective in a series of in-depth experiments. Finally, our CISM achieved the new state-of-the-art results on the challenging iSAID [20] dataset only with image-level labels. To achieve a comparison with more competitive algorithms, we compared the results of our methods with existing NS methods on the PASCAL VOC2012 [21] dataset, and the mIoU results on the validation and test sets were 67.3% and 68.5%, respectively. Following is a brief summary of the main contributions:

1.  We propose CISM as the first RS cross-image semantic mining WSSS framework to explore more object regions and complete semantics in multi-category RS scenes with two novel loss functions: the common semantic mining loss and the non-common semantic contrastive loss, obtaining cutting-edge state-of-the-art results on the iSAID dataset by only using image-level labels.
2.  To capture cross-image semantic similarities and differences between categories, including the background, the prototype vectors based on class-specific feature maps and the PIE module are proposed, which are beneficial to enhance the feature representation and improve the target localization performance by 4.5% and 2.1% on the mIoU and OA, respectively.
3.  To avoid class confusion and the unnecessary activation of closely related backgrounds, we propose integrating the SLSC task and its corresponding novel loss into our framework, which forces the network to acquire knowledge from additional object parts while suppressing false activation regions. Ultimately, SLSC improved the model performance by 1.3% and 1.5% on the mIoU and OA, respectively.
4.  The multi-category sample generation strategy is proposed to obtain a large number of multi-category samples by randomly stitching most of the low-category samples, which balances the distribution of samples across different categories and improved the model performance by 1.9% and 2.5% on the mIoU and OA, respectively.

## 2. Related Works

### 2.1. Semantic Segmentation in Remote Sensing Scenes

Deep learning offers a powerful feature representation capability, making it possible to utilize fully supervised methods to segment high-resolution RS images with high accuracy. These methods, however, rely on time-consuming and laborious pixel-level annotations, severely limiting their application scenarios. Consequently, semi-supervised, weakly supervised, and unsupervised segmentation methods are steadily emerging in RS segmentation.

The majority of currently available semi-supervised methods [7,22,23] employ Generative Adversarial Networks (GANs) trained on datasets with few pixel-level annotations to achieve model performance as close to fully supervised segmentation as possible. Sun et al. [24] proposed a semi-supervised semantic segmentation framework with boundary awareness to infer pseudo-labels from very high-resolution unlabeled RS images using GANs with few annotations at the pixel level. He et al. [25] devised a semi-supervised learning strategy with consistent regularization and a hybrid perturbation paradigm to enhance the performance of models on land cover datasets. To further alleviate the training cost, some researchers [26,27] have started to investigate challenging unsupervised RS segmentation methods. Using topology learning, Grozavu et al. [26] proposed a method for automatically segmenting images into different regions according to the texture, intensity, and color of the images. In a study conducted by Scheibenreif et al. [27], large-scale RS data were pre-trained using the Transformer architecture before the model was fine-tuned on small labeled datasets. Nevertheless, without any human-supplied supervised input, unsupervised models frequently achieve very limited segmentation performance. As a compromise between fully supervised and unsupervised segmentation, Weakly Supervised Semantic Segmentation (WSSS) approaches give RS semantic segmentation a glimmer of hope. Related research on WSSS will be introduced in detail in Section 2.2.

### 2.2. Weakly Supervised Semantic Segmentation

Weakly Supervised Semantic Segmentation (WSSS) models aim to simplify annotation while minimizing accuracy loss in the semantic segmentation task. Existing WSSS annotation types have gradually diminished from pixel-level masks to points [28], scribbles [5,29], bounding boxes [6–9], and image-level labels [10–13]. Among these, image-level annotations present the most-challenging task due to only providing category information without location and shape information. Our approach focuses on WSSS approaches based on the most-challenging image-level annotations in RS scenes.

Currently available state-of-the-art methods follow the mainstream pipeline of WSSS, firstly training a classifier to generate the CAMs [14] as the initial seeds and further mapped into pseudo-masks. Then, segmentation networks with pseudo-masks as the ground truth labels are trained in fully supervised manner. However, existing methods generate CAMs, which only locate the most-discriminative regions of objects, and a large number of false activations are seen in the background. The focus of much existing research in this area is on improving the consistency and integrality of activation regions in CAMs. Specifically, Wei et al. [13] employed the antagonistic erasure strategy to gradually mine the perfect target region by an iterative process. Afterwards, Wei et al. [30] introduced the dilated convolution with varying dilation rates to expand the receptive field of the classifier and combined the contextual information of the target region to generate dense target localization. Wang et al. [18] incorporated the equivariant constraint to enable the consistency of CAMs under different image scale transformations and combined a self-supervised approach to improve the CAM quality. Chang et al. [31] integrate the self-supervised sub-category exploration into the classifier to force the network to acquire more comprehensive target features and improve the accuracy of segmentation. Lee et al. [32] proposed the FickleNet to combine diverse locations on feature maps of the convolutional neural network by the dropout layer, in order to mine more complete target regions.

Other work has been performed, such as adding additional saliency maps [33,34], other web-sourced images [35,36], or video data [37,38] to refine the generated initial CAM.

However, each of the above methods treats images independently of the others, ignoring the relationship in the cross-image context. Some cross-image semantic mining methods have been proposed to explore more consistent representations within the same class. Fan et al. proposed the CIAN [39] to create a pixel-level relationship matrix between different images of the same category and obtained mutually complementary information from different images.Another similar approach was proposed by Sun et al. [40], which designs a special classification network with co-attention and contrastive co-attention to discover integral object regions by addressing cross-image semantics. In addition, RPNet [41] was proposed to explore the cross-image object diversity of the training set, which identifies similar object parts in an image by region feature comparison and propagates object confidence between regions to discover new object regions while suppressing background regions.

Compared with the dataset of natural scenes, the background of RS images is complex and classified; hence, there are fewer studies for WSSS tasks in RS scenes. Fu et al. [42] proposed the WSF-NET with feature fusion and designed a binary training strategy on water and cloud datasets to eliminate the class imbalance problem. Chen et al. [43] proposed the SPMF-Net to identify the building regions by utilizing a top-down method and superpixel pooling to achieve accurate and complete segmentation results of buildings. Although the above methods are effective, they ignore the rich relationships across different images and only deal with no more than two simple categories. When faced with RS images containing multiple categories, the above work does not provide sufficient information for training a qualified segmentation model, with the segmentation results on multi-category RS datasets being poor. The present work proposes an innovative cross-image semantic mining framework to discover more object regions and complete the semantics for target localization and category differentiation in multi-category RS scenes. Meanwhile, extensive experiments confirmed that our method resolves some ambiguities and prevents false predictions by connecting correlated regions across images.

## 3. Methodology

To begin with, we discuss in greater detail the proposed CISM network structure, as shown in Figure 1. Then, we review the conventional CAM pipeline and its limitations. Based on this, we rethink the CAM generation process. Further, we introduce how the MCSG strategy balances the distribution of samples across different categories. Finally, we mainly introduce all the loss functions used in CISM.

### 3.1. Cross-Image Semantic Mining Network

As RS image scenes contain multiple categories, it is challenging to locate the target and differentiate between the categories. Existing algorithms only employ information from a single image to localize the target, neglecting the rich semantic context associated with weakly annotated data. To maximize the potential of the diverse training data, we propose CISM, a novel WSSS framework with a cross-image semantic mining capability, which is used to explore cross-image semantics in multi-category RS scenes. As shown in Figure 1, object features are extracted as prototype vectors from pairs of images containing the same object category, which is referred to as the query and reference image. By employing a two-branch Siamese network as opposed to the conventional single-branch Siamese network, complementary information from different images can be used to make more accurate and consistent estimates, and all of the data can be utilized to their full potential.

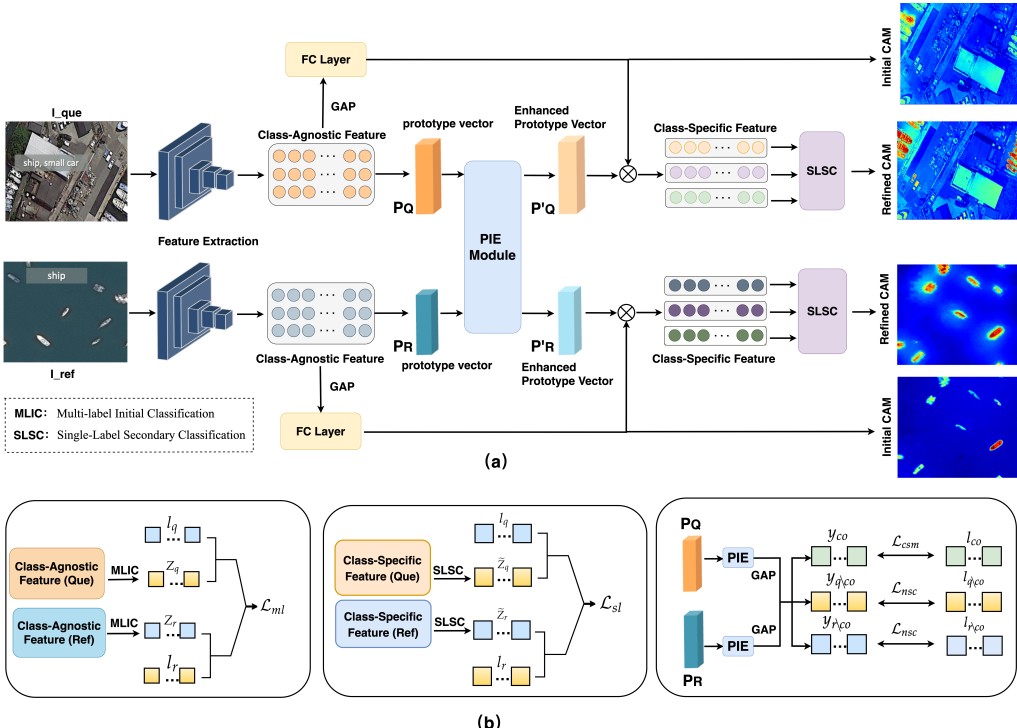

**Figure 1.** (**a**) An overview diagram of the WSSS cross-image semantic mining (CISM) framework. For a pair of query and reference images containing common classes as the network input, then enhancing the feature representation by exploring prototype vectors to generate class-specific features. (**b**) The schematic diagram of the all loss functions. The basic multi-label soft margin classification loss $\mathcal{L}_{ml}$ is used to provide the rough localization of objects. $\mathcal{L}_{sl}$, $\mathcal{L}_{csm}$, and $\mathcal{L}_{nsc}$ are three novel losses to assist in the training of the network.

To begin with, given the image pairs $[I_Q, I_R]$ as the input, the paired image features are extracted by the backbone network for further enhancement of the prototype vectors and generation of class-specific features. In order to obtain CAMs, a weighting factor is applied to the class-specific features based on their contribution to the classification output. For CAM refinement, our approach follows common practice in that we used denseCR F [44] to improve the CAM quality, followed by argmax to generate pseudo-masks based on the refined CAMs. Furthermore, we followed the pipelines of [17,18] for semantic segmentation by using DeepLabV1 [45] to train the model with pseudo-masks and determine the final segmentation outcome. CISM can construct shared feature relationships between targets of the same class in different sample images, as well as differential feature relationships between a single target class and all other targets and backgrounds. Our model will eventually be able to generate refined CAMs and pseudo-labels for all samples once the classifier has been trained.

### 3.1.1. Prototype Vector Generation

Inspired by RPNet [41], we selected only the activation regions with high confidence as the prototype vectors. Unlike the RPNet, we implemented interactive enhancement based on prototype vectors generated by class-agnostic features, aiming to purposefully extract more common category features and strengthen the discriminability between different categories. The details of the interactive enhancement of prototype vectors will be introduced in Section 3.1.2. Specifically, given the image pair $[I_Q, I_R]$ as the input, the paired image features $F_n^q(x, y)$ and $F_n^r(x, y)$ are extracted by the backbone network.

Then, the initial prototype vectors are extracted by the channel-unified convolution layer of $1 \times 1$ with a learnable matrix $W$, and the sigmoid function $\sigma$ is directly applied after $W$ as

$$F_c^q(x,y) = \sigma(\mathcal{F}_{1\times1}(F_n^q(x,y))) = \sigma(W_k^\top F_n^q(x,y)) \tag{1}$$

$$F_c^r(x,y) = \sigma(\mathcal{F}_{1\times1}(F_n^r(x,y))) = \sigma(W_k^\top F_n^r(x,y)) \tag{2}$$

To ensure that only the activation region with high confidence is selected as the prototype vector, we designed a learnable gate map $\Gamma(x,y)$ to threshold the corresponding initial prototype vectors $F_c^q(x,y)$ and $F_c^r(x,y)$ with a learnable parameter $\eta$. The initial prototype vector $F_c(x,y)$ is then masked by the learnable gate map $\Gamma(x,y)$ to extract the updated prototype vector $P_Q \in \mathbb{R}^{1\times1\times C}$ and $P_R \in \mathbb{R}^{1\times1\times C}$. Specifically, we employed Masked Average Pooling (MAP) [46] for embedding different foreground objects in different initial prototype vectors $F_c(x,y)$, on which pixels within the object regions are averaged to obtain the updated prototype vectors as follows:

$$P_Q = \frac{\sum_{x,y} F_c^q(x,y)\Gamma(x,y)}{\sum_{x,y} \Gamma(x,y)} \tag{3}$$

$$P_R = \frac{\sum_{x,y} F_c^r(x,y)\Gamma(x,y)}{\sum_{x,y} \Gamma(x,y)} \tag{4}$$

where

$$\Gamma(x,y) = \Phi(\mathbf{1}^{H\times W}, \eta) \tag{5}$$

The parameter $\eta$ is an adjustable variable, which helps mask the corresponding locations of the initial prototype vector $F_c(x,y)$ by, during training, randomly discarding some activated regions from CAMs. $\Phi$ is the dropout function, which randomly removes some elements of the matrix $\mathbf{1}^{H\times W}$. Different foreground objects are embedded in different prototype vectors using MAP, which helps the network to learn object features better while ignoring the background and noise. An illustration of the complete process of prototype vector generation is shown in Figure 2.

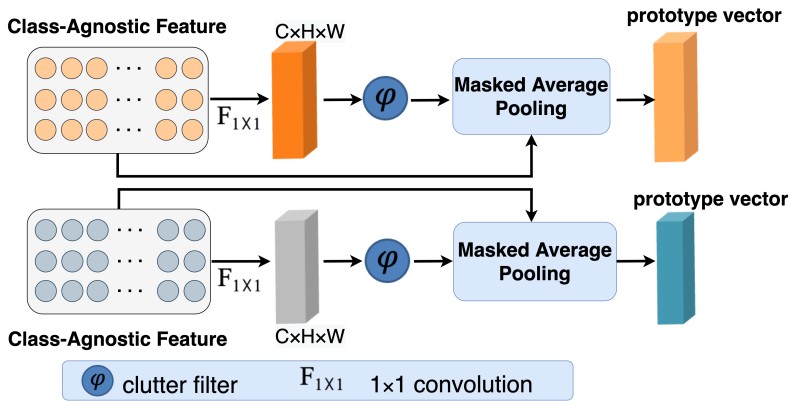

**Figure 2.** The schematic diagram of prototype vector generation.

### 3.1.2. Prototype Interactive Enhancement

As seen in the previous section, prototype vectors generated by existing mainstream methods still contain only the information of a single image, and the images in the datasets are independent of each other. When faced with the RS scenes containing multiple categories, the extraction of target features becomes incomplete and may also introduce other categories or background, which leads to the final target localization becoming more difficult. In our work, as shown in Figure 3, we propose a Prototype Interactive Enhancement

(PIE) module to integrate the common target features in the reference image into the prototypes of the query image. Similarly, the common target features in the query image would also be embedded in the reference image. The PIE module enhances the same category of features while also indirectly enhancing the discriminative nature of different categories of features. Specifically, we first define $P_Q$ and $P_R$ as the query prototypes and reference prototypes waiting for the input; $\mathcal{F}_{1\times1}$ denotes the $1 \times 1$ convolution operation; $X_Q, X_R$ denote the transformed query prototypes and reference prototypes, respectively. We have

$$X_Q = \mathcal{F}_{1\times1}(P_Q), \quad X_R = \mathcal{F}_{1\times1}(P_R) \tag{6}$$

where $\mathcal{F}_{1\times1}$ is mainly used here to change the shape of the prototypes, and the sizes of the query prototypes and reference prototypes are both $[C, H, W]$.

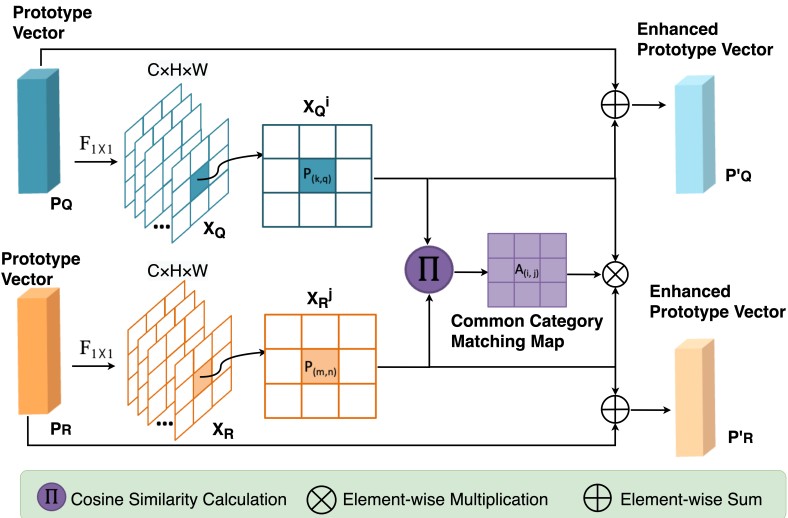

**Figure 3.** The schematic diagram of the PIE module.

In order to complete the embedding of the common category features, we designed a common category matching map $A_{(i,j)}$ that reveals the pixel-by-pixel correspondence between $X_Q$ and $X_R$. A high value in $A_{(i,j)}$ means that one pixel in the query feature $X_Q$ corresponds to at least one pixel in the reference feature $X_R$. To generate the common category matching map, the cosine similarity calculation $cos(X^q_{:,i,j}, X^r_{:,i,j}) \in \mathbb{R}^{H\times W}$ between feature vectors $X^q_{:,i,j} \in X_Q$ and $X^r_{:,i,j} \in X_R$ is first conducted at the pixel level. Then, we compiled all the maximum similarity values and created the common category matching maps as

$$A_{(i,j)} = \max_{i,j\in\{1,2,\dots,HW\}} \frac{(X^q_{:,i,j})^T (X^r_{:,i,j})}{\left\| X^q_{:,i,j} \right\| \left\| X^r_{:,i,j} \right\|} \tag{7}$$

where the similarity value of the pixel $(i, j)$ is denoted by $A_{(i,j)}$. To obtain enhanced prototype vectors, the common category matching map $A_{(i,j)}$ is utilized to extract similar parts of the features between reference images and query images. Note that before extraction, we normalized the value of $A_{(i,j)}$ to be in the interval 0–1. Firstly, we extracted similar features from the transformed prototypes $X_Q$ and $X_R$ by the normalized $A_{(i,j)}$. Then, all messages about similar parts from query features and reference features are obtained in a complementary way. Specifically, we found similar target features in the same category as query feature $X_Q$ from reference feature $X_R$, and the same steps are followed in query feature $X_Q$. Finally, the similar target features extracted by $X_Q$ and $X_R$ from each other are merged into the original prototype vectors $P_Q$ and $P_R$. The enhanced prototype vectors will

be sent to generate class-specific features. The entire generation process of the enhanced prototype vectors is represented as:

$$P'_Q(i,j) = P_Q(i,j) + \left[ \frac{\sum_{i,j} A_{(i,j)}(X^r_{:,i,j})}{\sum_{i,j} A_{(i,j)}} \right] \tag{8}$$

$$P'_R(i,j) = P_R(i,j) + \left[ \frac{\sum_{i,j} A_{(i,j)}(X^q_{:,i,j})}{\sum_{i,j} A_{(i,j)}} \right] \tag{9}$$

where $P'_Q$ and $P'_R$ stand for the enhanced query prototypes and reference prototypes, respectively. Consequently, the common category matching map incorporates the features of both the reference and the query images and achieves mutual embedding of reference prototypes and query prototypes.

In this part, the key to the proposed prototype interactive enhancement method is to produce the common category matching map $A_{(i,j)}$ by extracting the maximum value from the similarity matrix using the query image and the reference image, as shown in Equation (7).

### 3.2. Revisiting the CAM Generation

Most previous WSSS research mainly relied on Class Activation Maps (CAMs) to generate pixel-level pseudo-labels for fully supervised training. The first part of this section discusses the conventional CAM pipeline and its limitations. To enhance the integrity and fine granularity of CAMs, we further explored the CAM generation process and propose appropriate improvement methods.

### 3.2.1. Multi-Label Initial Classification

With an input image $I \in \mathbb{R}^{3 \times H \times W}$ and corresponding image-level label $Y \in \mathbb{R}^{1 \times C}$ as the inputs, initially, a multi-label classification model is trained based on Global Average Pooling (GAP), and the following Fully Connected (FC) layer is then applied to make predictions. Thus, the results of the prediction can be expressed as

$$Z_q = \text{FC}(\text{GAP}(F^q_n(x,y))) \tag{10}$$

$$Z_r = \text{FC}(\text{GAP}(F^r_n(x,y))) \tag{11}$$

where the paired image features $F^q_n(x,y)$ and $F^r_n(x,y)$ are extracted by the backbone network from the input images $I_Q$ and $I_R$. $Z_q$ and $Z_r$ are the prediction scores obtained by the classifier. $C$ and $H \times W$ correspond to the channels and spatial dimensions, respectively. The CAM is extracted based on the contributions of image features $F_n(x,y)$ in each category. For brevity, we denote the CAM as $M_c(x,y) \in \mathbb{R}^{C \times H \times W}$ as follows:

$$M^q_c(x,y) = \Theta\left( \frac{W_c^T \odot F^q_n(x,y)}{\max(W_c^T \odot F^q_n(x,y))} \right) \tag{12}$$

$$M^r_c(x,y) = \Theta\left( \frac{W_c^T \odot F^r_n(x,y)}{\max(W_c^T \odot F^r_n(x,y))} \right) \tag{13}$$

where $W_c$ denotes the classifier weight for the $c$-th class and $F^q_n(x,y)$ and $F^r_n(x,y)$ represent the feature maps of input image $I_Q$ and $I_R$ prior to the GAP. In addition, $\odot$ is the element-wise multiplication, while $\Theta$ is the activation function, which in our experiment was *ReLU*.

### 3.2.2. Single-Label Secondary Classification

In the above process of generating CAMs, the GAP layer is employed to obtain the prediction vector $Z_q \in \mathbb{R}^{C \times 1 \times 1}$ and $Z_r \in \mathbb{R}^{C \times 1 \times 1}$. Due to the fact that all pixels have basically been assigned the same class label, it is difficult for traditional classifiers to ensure

no reduction of positive regions while penalizing negative regions and no expansion of negative regions while encouraging positive regions. The resultant consequences are category confusion in multi-category cases and the inappropriate activation of related backgrounds. To alleviate the above problem, we extracted class-specific features based on the above multi-label initial classification. The specific extraction process is as follows:

$$\widetilde{F}_n^q(x,y) = (P'_Q \otimes \mathrm{M}_c^q(x,y)) \odot F_n^q(x,y) \tag{14}$$

$$\widetilde{F}_n^r(x,y) = (P'_R \otimes \mathrm{M}_c^r(x,y)) \odot F_n^r(x,y) \tag{15}$$

where $\otimes$ denotes the vector multiplication of $M_c(x,y)$ and the enhanced prototype vector $P$, which integrate the cross-image semantic similarities and differences between categories (including background) into $M_c$. The $\odot$, as the elementwise multiplication, is then applied to the class-specific features $\widetilde{F}_n$. In addition, we adjust the process of generating class scores to enforce the network to learn from more object parts. As shown in Figure 4, the pixel confidence map $\gamma_{c,i,j} \in \mathbb{R}^{C \times H \times W}$ is obtained by the softmax function instead of the GAP layer, which provides the ability to adaptively assign a soft label to each pixel. In order to calculate the final class score $\widetilde{Z}$, we calculated the channel-wise weighted average of the class-specific features $\widetilde{F}_n(x,y)$, treating the pixel confidence map $\gamma_{c,i,j}$ as the weight .

$$\widetilde{Z}_q = \frac{\sum\limits_{(i,j)\in R} \gamma_{c,i,j} \times \widetilde{F}_n^q(x,y)}{\sum\limits_{(i',j')\in R} \gamma_{c,i',j'}} \tag{16}$$

$$\widetilde{Z}_r = \frac{\sum\limits_{(i,j)\in R} \gamma_{c,i,j} \times \widetilde{F}_n^r(x,y)}{\sum\limits_{(i',j')\in R} \gamma_{c,i',j'}} \tag{17}$$

$\widetilde{Z}_q$ and $\widetilde{Z}_r$ would be sent to calculate the single-label secondary classification loss, penalizing misclassified pixels and encouraging the network to locate more target regions. The proposed method filters class-agnostic features by CAMs, which facilitates the activation of positive categories and inhibits the false activation of negative categories in the classification.

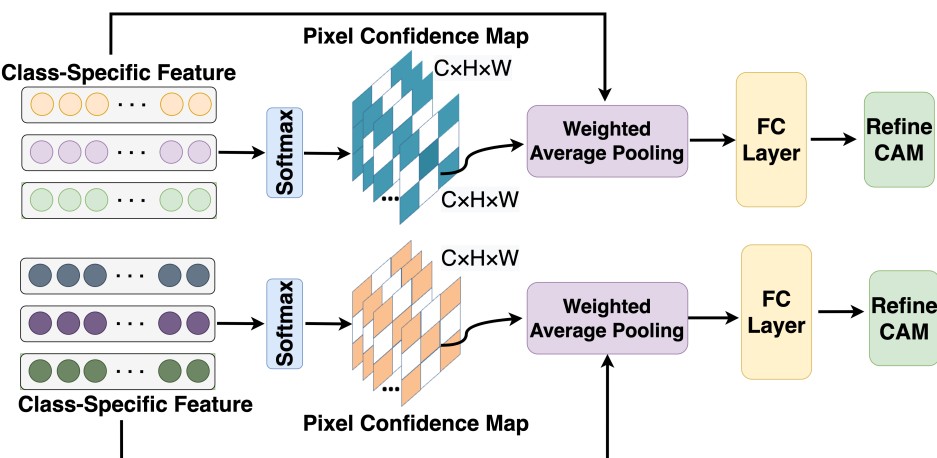

**Figure 4.** The schematic diagram of SLSC. The soft labels are adaptively assigned to each pixel with the softmax operation.

### *3.3. Multi-Category Sample Generation*

As shown in Figure 5, PASCAL VOC2012 contains 64% single-category scenes, whereas RS datasets contain fewer than 30% single-category scenes. In proportion to the increase in categories, the number of samples plummets, especially in the PASCAL VOC2012 dataset, where less than 10% of the samples are from three or more categories and almost none are from five or more categories. The increase in the number of categories greatly enhances the difficulty of the classification and final segmentation tasks, and the plummeting number of samples further escalates the challenge, which is clearly reflected in the RS dataset represented by iSAID. In the iSAID dataset, 63% of the samples contain targets from 2–3 categories, and samples with more than 4 categories account for 9% of the samples. We summarize the above discussion as the problem of "uneven distribution of multi-category samples", which leads to a sudden drop in the performance of existing models.

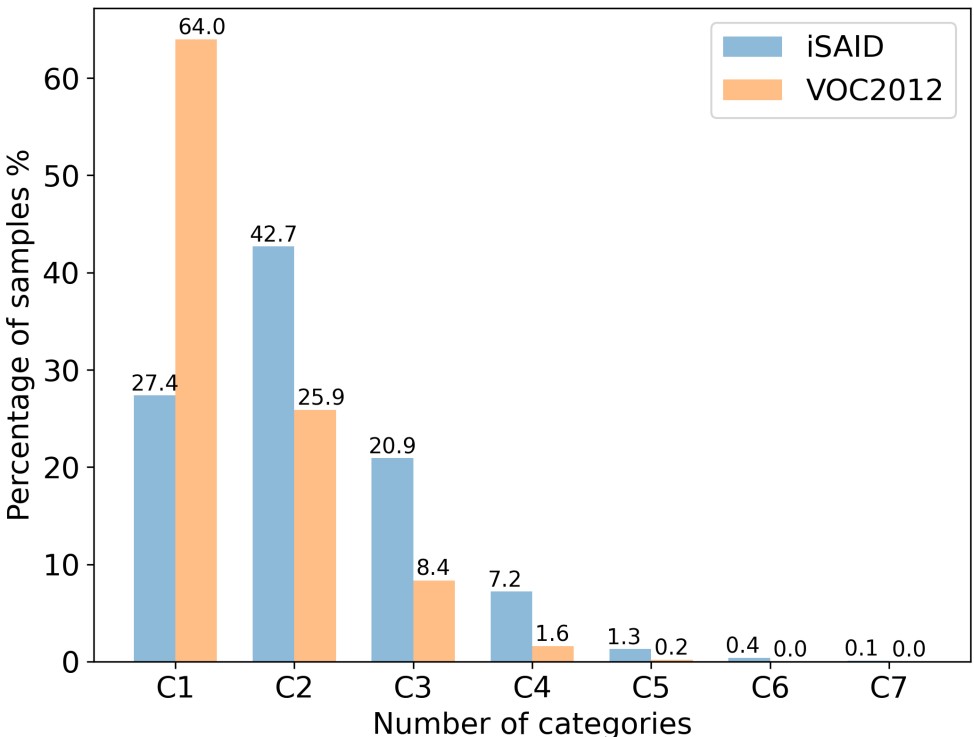

**Figure 5.** The proportion of images with various scene categories in the RS and NS datasets, which represents the number of images with various scene categories in relation to all images in the datasets.

This paper proposes a method for enhancing data for MCSG by randomly splicing low-category samples with different numbers of categories to boost the number of samples comprising multiple categories. In particular, we present the classification of images with the number of categories in iSAID below 3 (including 3) as the low-category sample group $\{S_i^L | i = 1, ..., n_L\}$ and the images with category numbers above 3 as the high-category sample group $\{S_i^H | i = 1, ..., n_H\}$, where $n_L$ and $n_H$ represent the number of samples in the low-category sample group and high-category sample group, respectively. There are roughly three steps involved in the data manipulation process:

- First, we randomly selected images from the $\{S_i^L | i = 1, ..., n_L\}$ and $\{S_i^H | i = 1, ..., n_H\}$ groups to form a batch denoted as $B\{S_i^L, S_i^H\}$ according to the ratio $\lambda$, where $b$ is the size of the batch size ($b \geq 4$), and the ratio $\lambda = n_L / (n_H + n_L)$.
- Second, four images were randomly selected from the $Random(S_i^L | i = 1, ..., \lambda b)$. *Random* denotes the randomly selected operation.

- Third, the selected images $\{S_i^L \| i = 1, 2, 3, 4\}$ were stitched into a new image according to random scaling, cropping, and random arrangement. In particular, the new image was the same size as the maximum size of the selected images.

The structure diagram of the MCSG strategy is shown in Figure 6. In the MCSG module, a large number of low-category samples are stitched in a random scale, random crop, and random arrangement, which drastically increases the number of samples from the high-category groups. Although the process is simple, the MCSG module balances the sample distribution across the various categories and vastly enhances the diversity of images.

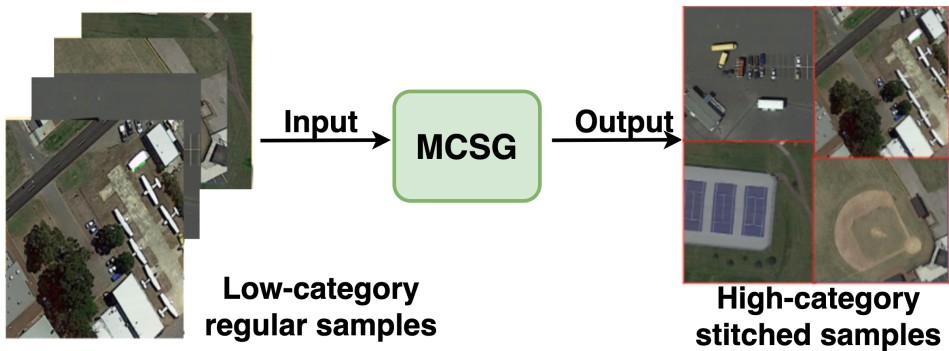

**Figure 6.** The schematic diagram of MCSG. The low-category samples are stitched in a random scale, random crop, and random arrangement in order to produce more high-category samples.

### 3.4. Network Training Loss

Detailed descriptions of the overall loss functions of our CISM framework are presented in this section. The overall loss $\mathcal{L}$ is expressed as

$$\mathcal{L} = \mathcal{L}_{ml} + \lambda(\mathcal{L}_{sl} + \mathcal{L}_{csm} + \mathcal{L}_{nsc}) \tag{18}$$

where the basic multi-label soft margin classification loss $\mathcal{L}_{ml}$ is used to provide the rough localization of objects. $\mathcal{L}_{sl}$ denotes as single-label secondary classification loss, which is utilized to compensate the sparsity of the initial CAM. $\lambda$ is a learnable parameter for balancing the losses. On the basis of a double-branch Siamese network, we propose the Common Semantic Mining (CSM) loss $\mathcal{L}_{csm}$ and the Non-common Semantic Contrastive (NSC) loss $\mathcal{L}_{nsc}$, which are primarily used for compelling the network to acquire common category features and improve the robustness of discriminating between non-common categories.

#### 3.4.1. Multi-Label Initial Classification

Since the training phase provides only image-level category labels, the first step in training the network is to train a model for multi-label classification using the multi-label soft margin classification loss $\mathcal{L}_{ml}$, according to previous research [16,17]. $\mathcal{L}_{ml}$ is defined as follows:

$$\mathcal{L}_{ml}^q = -\sum_{c=1}^{C} l_q \cdot \log \sigma(Z_q) + (1 - l_q) \cdot \log(1 - \sigma(Z_q)) \tag{19}$$

$$\mathcal{L}_{ml}^r = -\sum_{c=1}^{C} l_r \cdot \log \sigma(Z_r) + (1 - l_r) \cdot \log(1 - \sigma(Z_r)) \tag{20}$$

where $\sigma(\cdot)$ is the sigmoid activation function and $l_q \in \{0, 1\}^C$ and $l_r \in \{0, 1\}^C$ are image labels for C semantic categories in the query image $I_Q$ and the reference image $I_R$. $Z_q \in$

$\mathbb{R}^{C \times 1 \times 1}$ and $Z_r \in \mathbb{R}^{C \times 1 \times 1}$ are the corresponding prediction vectors of image classification. The final $L_{ml}$ is the average of $L_{ml}^q$ and $L_{ml}^r$.

### 3.4.2. Single-Label Secondary Classification

To avoid class confusion and unnecessary activation of closely related backgrounds, the initial CAM serves as a mask in order to extract class-specific features from the class-agnostic ones obtained by the previous multi-label classification process. As can be seen from the above multi-label classification process, the GAP layer is employed to obtain the prediction vector $Z_q \in \mathbb{R}^{C \times 1 \times 1}$ and $Z_r \in \mathbb{R}^{C \times 1 \times 1}$, in which all pixels have basically been assigned the same class label. Unlike the traditional equal-weighted average, we remove the GAP layer and choose the softmax function to adaptively assign soft labels to each pixel. Specifically, the class-specific features are weighted by the confidence map $\gamma_{c,i,j}$ for the calculation of the average channel-wise weight and predict the final class scores $\widetilde{Z}$. Inspired by ReCAM [47], we design the single-label secondary classification loss $\mathcal{L}_{sl}$ to penalize any misclassification caused by either poor features or poor masks. The following is the definition of $\mathcal{L}_{sl}$:

$$\mathcal{L}_{sl}^q = -\frac{1}{\sum_{i=1}^N l[i]} \sum_{n=1}^N l[n] \log \frac{\exp(\widetilde{Z}_q[n])}{\sum_j \exp(\widetilde{Z}_q[j])} \tag{21}$$

$$\mathcal{L}_{sl}^r = -\frac{1}{\sum_{i=1}^N l[i]} \sum_{n=1}^N l[n] \log \frac{\exp(\widetilde{Z}_r[n])}{\sum_j \exp(\widetilde{Z}_r[j])} \tag{22}$$

where $l[n]$ and $\widetilde{Z}[n]$ are, respectively, the $n$-th elements of $l$ and $\widetilde{Z}$, and the final $L_{sl}$ is the average of $L_{sl}^q$ and $L_{sl}^r$.

### 3.4.3. Common Semantic Mining Loss

To facilitate object pattern mining, we utilized the common class labels as supervised signals and extracted common features from the transformed query prototypes $X_Q$ and reference prototypes $X_R$ with the common category matching map $A_{(i,j)}$ to develop the common prediction vector $y_{co} \in \mathbb{R}^{C \times 1 \times 1}$, respectively. The generation process of $y_{co}$ is expressed as:

$$y_{co} = \text{GAP}(\Psi(A_{(i,j)} \times X_Q)) + \text{GAP}(\Psi(A_{(i,j)} \times X_R)) \tag{23}$$

where $\Psi$ is the linear transform operation, implemented by $1 \times 1$ convolution layers. Then, we further applied the GAP layer to obtain the common prediction vector $y_{co} \in \mathbb{R}^{C \times 1 \times 1}$ for $I_Q$ and $I_R$, respectively. Then, $y_{co}$ would be utilized to compute the common semantic mining loss $\mathcal{L}_{csm}$ to assist in training the classifier, resulting in the discovery of relevant semantics in features and the recognition of common category objects. Intuitively, the loss $\mathcal{L}_{csm}$ between the query and reference images for common semantic classification is defined as follows:

$$\mathcal{L}_{csm} = -\sum_{c=1}^C l_{co} \cdot \log \sigma(y_{co}) + (1 - l_{co}) \cdot \log(1 - \sigma(y_{co})) \tag{24}$$

where the common semantic label $l_{co}$ is obtained from the ground truth labels $[l_q, l_r]$ of the paired images. The common semantic mining loss $\mathcal{L}_{csm}$ explicitly assists the classifier in tying semantic labels to corresponding object regions within common categories. At the same time, $\mathcal{L}_{csm}$ also facilitates the classifier to be aware of the relationship between the various components of objects. In essence, our method comprehensively utilizes context across almost all training data.

3.4.4. Non-Common Semantic Contrastive Loss

In addition to the preceding operation, which explores common semantics across images, we propose a non-common semantic contrastive loss $\mathcal{L}_{nsc}$ to mine complementary information based on differences in semantic content among paired images. By filtering the common category object information, we obtain the contrastive category object information in $I_Q$ and $I_R$, allowing the classifier to pay more attention to mining non-common semantics from the rest of the image regions. Intuitively, as in the above $y_{co}$ generation process, we obtain the corresponding non-common prediction vectors as follows:

$$y_{q \backslash co} = \text{GAP}(\Psi((\mathbf{1} - A_{(i,j)}) \times X_Q)) \tag{25}$$

$$y_{r \backslash co} = \text{GAP}(\Psi((\mathbf{1} - A_{(i,j)}) \times X_R)) \tag{26}$$

where $y_{q \backslash co}$ is the non-common prediction vector of the query image $I_Q$, but not of the reference image $I_R$, and $y_{r \backslash co}$ is the same. We applied the same procedure as in Section 3.4.3 to obtain the common prediction vector $y_{co} \in \mathbb{R}^{C \times 1 \times 1}$ to obtain the non-common prediction vectors $y_{q \backslash co}$ and $y_{r \backslash co}$. Then, the non-common semantic contrastive loss $\mathcal{L}_{nsc}$ is calculated utilizing the non-common prediction vectors $y_{q \backslash co}$ and $y_{r \backslash co}$, which allows the classifier to distinguish between different objects based on their semantics, as the contrastive co-attention disentangles the semantics of common and uncommon objects. The contrastive loss of non-common semantic classification can be represented as

$$\mathcal{L}_{nsc} = -\sum_{c=1}^{C} l_{q \backslash co} \cdot \log \sigma\left(y_{q \backslash co}\right) + \left(1 - l_{q \backslash co}\right) \cdot \log\left(1 - \sigma\left(y_{q \backslash co}\right)\right)$$

$$-\sum_{c=1}^{C} l_{r \backslash co} \cdot \log \sigma\left(y_{r \backslash co}\right) + \left(1 - l_{r \backslash co}\right) \cdot \log\left(1 - \sigma\left(y_{r \backslash co}\right)\right) \tag{27}$$

where the contrastive non-common semantic label $l_{q \backslash co}$ is derived from the removal of the $l_q$ of the common category label $l_{co}$, and the same goes for $l_{r \backslash co}$. By utilizing paired samples, we indirectly increased the total number of samples, bringing the number of training images closer to its square. The common category object information and the contrastive category object information make the classifier more aware of the semantics of the objects.

**4. Experimental Results and Discussion**

Numerous experiments were conducted on the challenging RS dataset iSAID [20] to evaluate the efficacy of key components of our algorithm. In addition, our analyses were also carried out on the PASCAL VOC2012 dataset in order to assess its performance in comparison with more state-of-the-art approaches. The detailed results of the experiment are discussed in the following paragraphs.

*4.1. Datasets and Evaluation Metrics*

As a public large-scale RS dataset, iSAID [20] was primarily obtained from the JL-1 and GF-2 satellites for the segmentation of aerial imagery, which has 15 target categories and one Background (BG) category. Specific categories include Basketball Courts (BCs), Baseball Diamonds (BDs), Bridges (BRs), Ground-Track Fields (GTFs), Harbors (HAs), Helicopters (HCs), Large Vehicles (LVs), Planes (PLs), Roundabouts (RAs), Soccer Ball Fields (SBFs), Ships (SHs), Swimming Pools (SPs), Storage Tanks (STs), Small Vehicles (SVs), and Tennis Courts (TCs). In lieu of the full names, the abbreviations in brackets were used in the results of our experiment. According to the initial partitioning of the iSAID dataset, 1411 high-resolution images were utilized for training and 458 for validation. In addition, each image was randomly sliced into 6–8 512 × 512 slices. As a result, 7500 slices of the training set, 1653 slices of the verification set, and 1315 slices of the test set were generated. There are 20 foreground categories in PASCAL VOC2012, as well as 1 background category,

which is commonly used to test the performance of object detection and image segmentation algorithms. Furthermore, PASCAL VOC2012 was augmented with images from the SBD [48] to obtain 10,582 images for the training phase, 1449 for the validation phase, and 1456 for the test phase. On the above two datasets, we only used image-level category labels to generate CAMs during training, which is the most-challenging task in WSSS. For the purposes of evaluating the quality of pseudo-labels, we utilized pixel-level ground truth labels in the inference stage. Furthermore, the standard mIoU was the metric applied to all of our experiments. Our comprehensive comparison of diverse datasets yielded more reasonable results in terms of performance.

### 4.2. Implementation Details

We adopted CIAN [39] as the baseline model in this paper. In the course of training, we employed the same data augmentation as in previous research [16,39,49,50]. Our network was trained with 20 iterations on a single NVIDIA RTX 3090 GPU using the Adam optimizer [51] with a batch size of 16. With the polynomial attenuation strategy [52], the learning rate decreased by 0.9 from the initial 0.01 level as each iteration completed. We followed previous research [16] by selecting ResNet-50 [53] as the backbone for the initial CAM generation. For a fair comparison, similar to the previous research, the dense CRF [44] was adopted as the post-processing procedure to refine the initial CAMs as the pseudo-labels. Finally, our segmentation model was trained with only pseudo-labels utilizing DeepLab [54] as the segmentation network.

### 4.3. Comparison with State-of-the-Art Results

This part examines the performance of CISM compared with the weakly supervised SOTA methods on the iSAID dataset and the PASCAL VOC2012 dataset. A more detailed explanation is provided below.

#### 4.3.1. iSAID Dataset

**Comparison with weak supervision methods:** For the purposes of comparing the performance with other WSSS methods, we selected several recent competitive NS methods to participate in this comparison. According to Table 1, for the performance comparison, we achieved state-of-the-art performances on the iSAID dataset without any additional training information. In comparison with the baseline method [39], our method improved the mIoU by 10.2%, which is a promising development in the field of WSSS. Notice that there are some very low values in Table 1, such as for SVs, HAs, HCs, etc. They have in common that they belong to hard samples, where the distribution of SVs is dense and the target scale is small, the target scale of HAs is highly variable, and the sample size of HCs is sparse. These hard samples are not conducive to network learning and are difficult to find in complex remote sensing scenes. The above experimental results demonstrated the advantage of leveraging cross-image relationships, which enables the implementation of image-level WSSS in the RS field.

**Table 1.** The performance statistics of weakly supervised methods in the mIoU (%) on iSAID.

| Model | BG | GTF | SBF | SV | SH | BR | BC | BD | RA | PL | TC | LV | ST | HA | SP | HC | mIoU |
|---|---|---|---|---|---|---|---|---|---|---|---|---|---|---|---|---|---|
| AffinityNet [17] | 77.8 | 14.7 | 43.7 | 3.7 | 5.8 | 6.3 | 18.8 | 20.4 | 9.8 | 7.5 | 54.3 | 23.8 | 42.4 | 3.9 | 1.8 | 1.0 | 20.9 |
| IRNet [16] | 78.6 | 18.9 | 39.8 | 6.7 | 16.6 | 7.5 | 24.5 | 16.4 | 5.7 | 15.7 | **54.9** | 29.0 | 29.6 | 9.8 | 2.7 | 3.1 | 22.5 |
| CIAN [39] | 78.3 | 21.9 | 40.8 | 8.9 | 17.5 | 8.3 | 28.4 | 22.7 | 13.5 | 18.1 | 47.5 | 27.6 | 30.2 | 8.5 | 4.2 | 5.3 | 23.9 |
| SEAM [18] | 73.2 | 28.4 | 37.8 | 4.7 | 15.3 | 6.9 | 43.1 | 31.4 | 16.7 | 19.6 | 44.3 | 31.1 | 24.4 | 7.8 | 9.8 | 8.3 | 24.5 |
| SC-CAM [31] | 76.2 | 28.6 | 36.1 | 11.1 | **20.0** | 5.7 | 23.1 | **36.9** | 23.2 | 24.8 | 54.3 | 30.1 | 29.8 | 10.3 | 17.3 | 11.9 | 26.6 |
| RPNet [41] | 80.2 | 33.6 | 42.1 | 9.8 | 16.9 | 7.2 | 45.2 | 32.6 | 26.1 | 27.6 | 51.5 | 29.4 | 42.7 | 9.3 | 16.8 | 13.7 | 30.3 |
| **CISM (ours)** | **83.1** | **35.5** | **44.6** | **11.8** | 19.5 | **8.4** | **48.3** | 34.4 | **29.2** | **30.5** | 52.6 | **31.9** | **44.1** | **10.8** | **17.6** | **14.3** | **32.2** |

**Comparison with other supervised methods:** To more intuitively compare the performance of the models under different supervision types, we compared four types of supervision methods and list their differences in Table 2, where $\mathcal{F}$ indicates full supervision,

$\mathcal{S}$ denotes semi-supervision, $\mathcal{W}(\mathcal{B})$ and $\mathcal{W}(\mathcal{I})$ denote the bounding box supervision and image-level supervision, respectively. Our CISM with ResNet50 achieved an mIoU of 34.1% on the iSAID dataset, outperforming all previous results only with image-level labels. However, compared to fully supervised and semi-supervised methods, our method had a large performance gap. The main reason is that the image-level supervision only provides information on target categories, while information on the number and location of targets is completely absent. Therefore, the bounding box supervision methods are proposed, which provide information on the categories and locations of all objects in the form of bounding boxes. Notably, our model was comparable to some bounding box works, such as [6,55,56], which were equal to 66.5% of the performance of the weakly supervised method [55] based on the bounding boxes. All the above experimental results demonstrated the advantage of leveraging cross-image relationships, which enables the implementation of image-level WSSS in the RS field.

**Table 2.** Evaluation results on the iSAID dataset under the different supervision types. We use different symbols to distinguish the type of supervision: $\mathcal{F}$ indicates full supervision. $\mathcal{S}$ indicates semi-supervision. $\mathcal{W}$ indicates weak supervision, where $\mathcal{W}(\mathcal{B})$ denotes bounding box supervision and $\mathcal{W}(\mathcal{I})$ denotes image-level supervision.

| Supervision Type | Model | Backbone | mIoU (%) |
|:---:|:---:|:---:|:---:|
| $\mathcal{F}$ | DeepLab v3 [54] | ResNet50 | 59.1 |
| $\mathcal{S}$ | SDI [55] | VGG16 | 54.9 |
| | Song [6] | VGG16 | 55.2 |
| | JMLNet [56] | ResNet50 | 56.8 |
| $\mathcal{W}$ (B) | SDI [55] | VGG16 | 53.8 |
| | Song [6] | VGG16 | 54.2 |
| | JMLNet [56] | ResNet50 | 55.3 |
| $\mathcal{W}$ (I) | CIAN [39] | ResNet50 | 23.9 |
| | SC-CAM [31] | ResNet50 | 26.6 |
| | RPNet [41] | ResNet50 | 30.3 |
| | **CISM (ours)** | ResNet50 | **35.8** |

**Analyses of the visualization:** Figure 7 illustrates the comparison between our method and the baseline method in terms of qualitative segmentation results on iSAID. From the first two rows, the CISM model can produce segmentation results with more explicit boundaries and more regular shapes, as well as significantly reducing misclassified pixels. The results in the middle two rows show that our method is able to reduce false positive predictions in the background to some extent, as they are never matched in any of the reference images. From the last two rows, we show that our CISM model can accurately recognize and locate objects in multi-category RS scenes. This is because our CISM builds a more robust representation across the dataset with the help of cross-image semantic information, and the related representations can be strengthened by each other, thus achieving more accurate semantic localization and segmentation. This is because our CISM constructs a more robust representation across the dataset using cross-image semantic information, and the related representations can be strengthened by each other, resulting in more accurate semantic localization and segmentation.

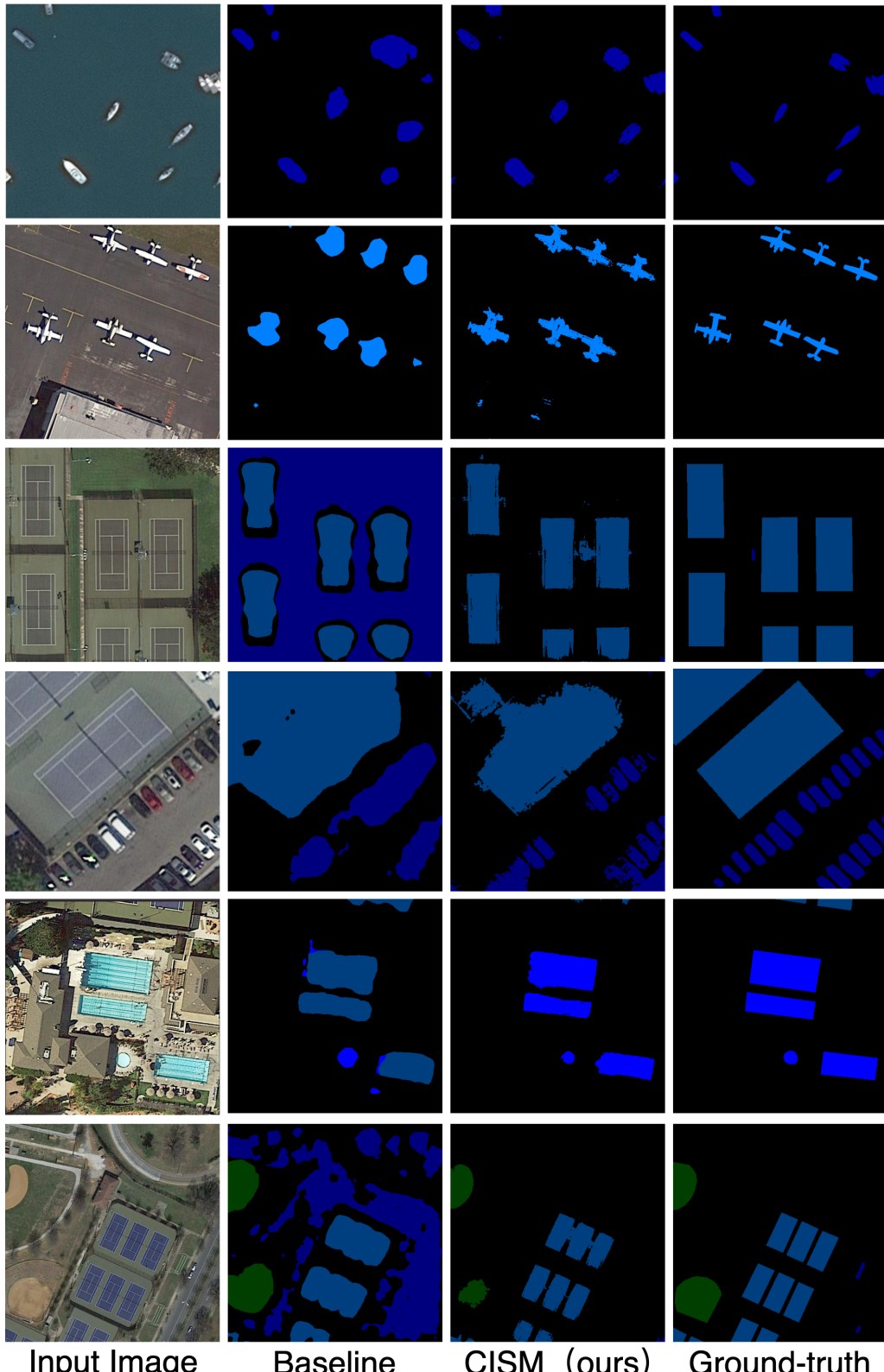

**Figure 7.** Visual illustration of the segmentation results of the iSAID dataset. The six rows show the results of the qualitative semantic segmentation to demonstrate how our CISM outperforms the baseline. Best viewed in color.

4.3.2. PASCAL VOC2012 Dataset

To achieve the comparison with more competitive algorithms, a further evaluation of the performance of the CISM model was conducted by applying it to the PASCAL VOC2012 dataset. As a standard convention, 10582 images were used for training, 1449 for validation, and 1456 for testing, with the mIoU serving as the metric for evaluating the segmentation results. The comparison results for the PASCAL VOC2012 are provided in greater detail below.

**Comparison with SOTA methods:** According to Table 3, our method produces better results than the baseline (CIAN) in both validation and testing and outperforms all of the previous results without any additional supervision. Based on the PASCAL VOC2012 validation set, our CISM achieved a 67.3% mIoU with ResNet-101. Furthermore, on the test set, we achieved a 68.5% mIoU and generated high-quality pseudo-masks. To be fair, the CISM model was also evaluated on various backbone networks and is marked with different combinations of the symbols * and †. Excluding the method that employs additional supervision, as a result of our model, all types of backbones achieved new state-of-the-art performance, with specific types including ResNet-38, ResNet-50, and ResNet-101, which demonstrates the effectiveness of cross-image semantic mining.

**Table 3.** The performance statistics of the current state-of-the-art on PASCAL VOC2012. We evaluated the CISM model with different backbone networks and signify these with various combinations of the symbols * and †. These different backbone networks include: ResNet38 (†); ResNet-50 († *); ResNet101 († **). The supervision types (Sup.) include: image-Level labels (*L*) and Saliency maps (*S*).

| Methods | Backbone | Sup. | Val | Test |
|---------|----------|------|-----|------|
| AFFNet [17] | R-38 | *L* | 61.7 | 63.7 |
| SEAM [18] | R-38 | *L* | 64.5 | 65.7 |
| A²GNN [57] | R-38 | *L* | 66.8 | 67.4 |
| EDAM [58] | R-38 | *L + S* | 70.9 | 70.6 |
| EPS [59] | R-38 | *L + S* | 70.9 | 70.8 |
| CIAN [39] | R-50 | *L* | 62.4 | 63.8 |
| IRNet [16] | R-50 | *L* | 63.5 | 64.8 |
| RPNet [41] | R-50 | *L* | 66.4 | 67.2 |
| MCOF [60] | R-101 | *L* | 60.3 | 61.2 |
| DCSP [61] | R-101 | *L* | 60.8 | 61.9 |
| DSRG [62] | R-101 | *L* | 61.4 | 63.2 |
| SeeNet [63] | R-101 | *L + S* | 63.1 | 62.8 |
| AISI [64] | R-101 | *L + S* | 63.6 | 64.5 |
| CIAN [39] | R-101 | *L* | 64.1 | 64.7 |
| FickleNet [32] | R-101 | *L + S* | 64.9 | 65.3 |
| SC-CAM [31] | R-101 | *L* | 66.1 | 65.9 |
| RPNet [39] | R-101 | *L* | 66.9 | 68.0 |
| *Ours:* | | | | |
| CISM (†) | R-38 | *L* | 64.4 | 66.8 |
| CISM († *) | R-50 | *L* | 66.8 | 67.6 |
| CISM († **) | R-101 | *L* | **67.3** | **68.5** |

*4.4. Ablation Experiments*

This section presents a series of thorough experiments designed to demonstrate the benefits of our CISM framework. To systematically assess the effectiveness of each CISM component, our evaluation of the CISM framework continued to be based on the CIAN [39] as the baseline and on the mIoU as the evaluation metric. On the more challenging iSAID dataset, all experimental results were evaluated.

### 4.4.1. Effectiveness of Prototype Vector

We obtained prototype vectors from class-specific features with the intent of selecting only highly confident activation regions to propagate and implement classification. As shown in Table 4, the addition of the prototype vector improved the mIoU by 0.7% compared to the baseline. As is observable, however, when we only considered the prototype vector to filter the low-confidence activation region, we obtained a small improvement in the mIoU of the CAMs, but a significant decrease in the OA of the CAMs, indicating that some of the activation area in the inner object regions was filtered out, resulting in a lower recall. Therefore, we propose the PIE module to achieve the complementary embedding of common target information in the reference and query images, aiming to compensate for the reduction of the target internal activation regions caused by the high-confidence filtering operation. With the addition of the PIE module, a further improvement of 2.8% for the mIoU was achieved on the prototype vector, and the OA value was directly increased to 79.3%. The above quantitative experimental results show the validity of the prototype vector and PIE module.

**Table 4.** The ablation research on the effectiveness of the prototype vector and the PIE module. We added the prototype vector and the PIE module sequentially to the baseline and observed the quality of the CAM on iSAID. +PV means the addition of the Prototype Vector, and +PIE means the addition of the PIE module. The mIoU (%) and OA (%) as the evaluation metrics for CAM quality are reported in the table.

| Baseline | +PV | +PIE | mIoU (%) | OA (%) |
|:---:|:---:|:---:|:---:|:---:|
| ✓ | | | 23.9 | 77.2 |
| ✓ | ✓ | | 24.6 | 73.5 |
| ✓ | ✓ | ✓ | 28.4 | 79.3 |

**Analyses of the visualization:** In order to illustrate the impact of the prototype vector and PIE module in an intuitive manner, the CAM visualization results are presented in Figure 8. Given a picture that includes some targets, the baseline can only have a rough estimate of the local discriminatory regions of the targets at first. Following the addition of the prototype vector and PIE module, there was a substantial improvement in the accuracy of target positioning and a strengthening of the CAM activation area, and the pseudo-masks became more complete and clearer. This is because that the prototype vector and PIE module suppress the further propagation of the background and interference information, ensuring that the classifier can learn more useful semantic information and achieve reasonable optimization.

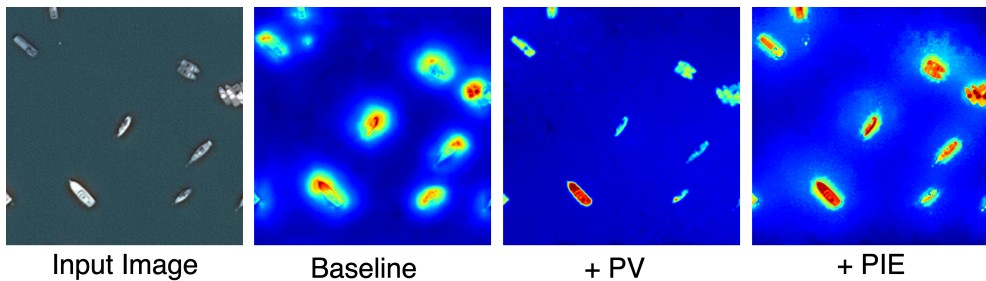

Input Image     Baseline     + PV     + PIE

**Figure 8.** The visualization after adding the prototype vector and PIE module.

**Selection of hyperparameters:** In the process of prototype vector generation, we developed an adjustable variable $\eta$ to assist in discarding certain regions from the CAMs during training, and then, the corresponding location of the initial prototype vector $F_c(x, y)$ will be masked. Table 5 presents the effect of different hyperparameter settings on CAM quality as measured by the mIoU (%) and OA (%). With the increase of the parameter $\eta$, the evaluation metrics gradually increased and then decreased, achieving the maximum value at $\eta = 0.3$.

**Table 5.** The effect of different hyperparameter settings on CAM quality as measured by the mIoU (%) and OA (%).

| $\eta$ | mIoU (%) | OA (%) |
|---|---|---|
| 0.1 | 28.9 | 80.6 |
| 0.3 | **32.2** | **82.4** |
| 0.5 | 27.4 | 80.1 |
| 0.7 | 25.8 | 79.2 |

4.4.2. Effects of Single-Label Secondary Classification

We appended the operation of Single-Label Secondary Classification (SLSC) to the initial multi-label classification on different networks to assess its validity independently. Here, we selected the CIAN network as the benchmark, along with the competitive RPNet network [39], which also employs the prototype vector, and our own CISM network for comparison. The mIoU and OA were also used as the evaluation metrics in our CISM framework to determine whether the SLSC is effective. We exploited the initial CAM to extract class-specific features in order to overcome class confusion and closely related background interference information. As indicated in Table 6, the addition of the SLSC increased the mIoU and OA by 3.8% and 3.3%, respectively, relative to the baseline. The middle two rows demonstrate that the addition of the SLSC improved the performance of the RPNet. As compared with the baseline, the mIoU improvement increased from 23.9% to 30.3% and the OA improvement rose from 77.2% to 80.8% after applying the SLSC, which further improved the two metrics to 32.6% and 82.7%, respectively. In the last two rows, we show the more significant gains of the SLSC on our CISM network. With the addition of the SLSC, we continued to improve above RPNet by a 0.9% mIoU and 1.2% OA. Noting that CISM alone can achieve higher performance than the RPNet highlights the benefit of the cross-image mining network.

**Table 6.** Ablation research of the effectiveness of the MCSG strategy. +SLSC means the addition of the Single-Label Secondary Classification. The effect of single-label secondary classification on the CAM quality using the mIoU (%) and OA (%) as the evaluation metrics.

| Baseline | +SLSC | mIoU (%) | OA (%) |
|---|---|---|---|
| ✓ | | 23.9 | 77.2 |
| ✓ | ✓ | 27.7 | 80.5 |
| **RPNet** | **+SLSC** | **mIoU(%)** | **OA(%)** |
| ✓ | | 30.3 | 80.8 |
| ✓ | ✓ | 32.6 | 82.7 |
| **CISM (ours)** | **+SLSC** | **mIoU (%)** | **OA (%)** |
| ✓ | | 32.2 | 82.4 |
| ✓ | ✓ | **33.5** | **83.9** |

**Analyses of the visualization:** In Figure 9, the CAM visualization results on the iSAID dataset are presented. The baseline network still has limited localization capability in multi-category RS scenes and is prone to produce incomplete activation areas. Compared to the baseline, the RPNet [39] found more activated regions by correlations between prototype vectors. However, the granularity of the CAM generated by the RPNet was still insufficient, hence limiting the quality of subsequent pseudo-label generation. Our CISM extracts prototype vectors from high-confidence regions while filtering out interference regions in the background and confusion regions between different categories, hence enhancing the representation of features. As can be seen in the last column of Figure 9, the addition of the SLSC improved the fine-grainedness of the CAMs and further refined the activation area in the CAMs.

| Baseline | RPNet | CISM | CISM + SLSC |

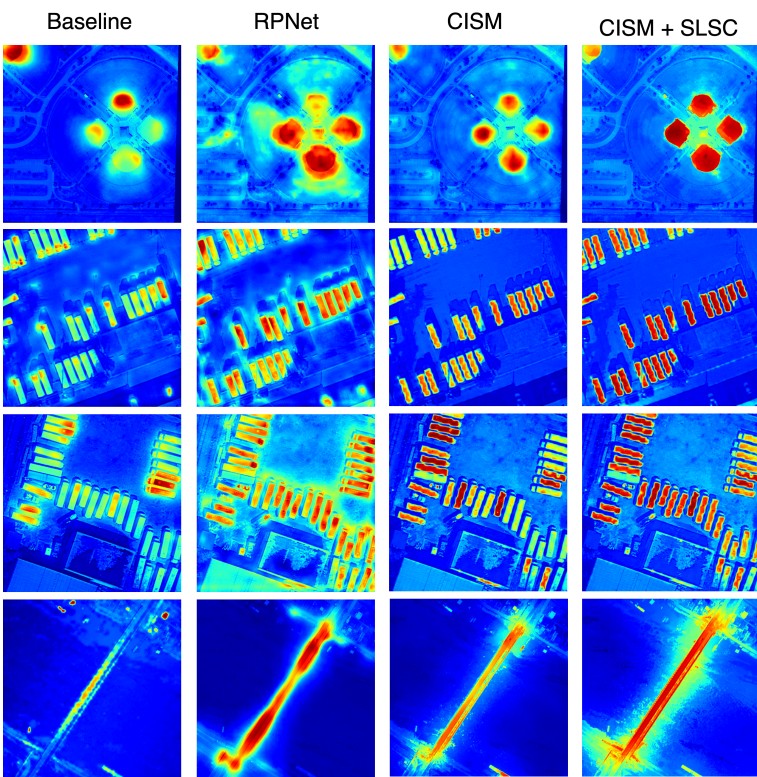

**Figure 9.** The visualization after adding the SLSC.

### 4.4.3. Effects of Multi-Category Sample Generation

To evaluate the Multi-Category Sample Generation (MCSG) strategy, we applied it to the baseline (CIAN) architecture, which is the first model to build cross-image relationships for WSSS and achieved the best results in the original NS studies. For the sake of equality, the hyperparameters and other conditions of our CISM model were identical to those employed by the CIAN architecture. Table 7 summarizes the comparison results. As we can see, there was a 1.9% mIoU improvement when the MCSG was applied to our CISM, compared to 1.4% on the baseline alone. Furthermore, we noticed that the baseline model still did not perform as well as our CISM model, even though the MCSG module was added. After applying the MCSG, our CISM improved the mIoU from 25.3% to 34.1%, which suggests that applying the MCSG greatly benefits the CAM quality.

**Table 7.** Ablation research of the effectiveness of the multi-category sample generation strategy. The mIoU (%) and OA (%) are the evaluation indicators of CAM quality.

| Method | mIoU (%) | OA (%) |
| --- | --- | --- |
| Baseline | 23.9 | 77.2 |
| Baseline + MCSG | 25.3 | 79.5 |
| CISM | 32.2 | 82.4 |
| CISM + MCSG | **34.1** | **84.9** |

### 4.4.4. Impact of Loss Functions

As explained in Section 3.4, our CISM was optimized through combining four loss functions, i.e., the basic multi-label soft-margin classification loss $\mathcal{L}_{ml}$, the single-label secondary classification loss $\mathcal{L}_{sl}$, the common semantic mining loss $\mathcal{L}_{csm}$, and the non-common semantic contrastive loss $\mathcal{L}_{nsc}$, which play different and important roles in guiding classification network training and generating CAMs. Here, we conducted related ablation experiments on iSAID for the purposes of further validating the effectiveness of each loss function. First, the quantitative comparison between different combinations of loss

functions is presented in Table 8. Our CISM only obtained a 23.9% mIoU when only $\mathcal{L}_{ml}$ was used. By removing the class-related background regions and the inter-class interference regions from the generated CAM, $\mathcal{L}_{sl}$ further improved the mIoU to 26.7%, which played an important role in enhancing the fine granularity of CAM. On the basis of this, in order to obtain more target features and improve the discriminative property between different targets, we extracted the common semantic mining loss $\mathcal{L}_{csm}$ and the non-common semantic contrastive loss $\mathcal{L}_{nsc}$ to provide guidance for network training. To independently examine the effect of $\mathcal{L}_{csm}$ and $\mathcal{L}_{nsc}$, we paired the $\mathcal{L}_{ml}$ loss and $\mathcal{L}_{sl}$ loss with $\mathcal{L}_{csm}$ and $\mathcal{L}_{nsc}$, respectively, to assist in supervising the training process for the network and then observed the CAM generation results. Note that the CAMs can obtain a 3.1% improvement in the mIoU when only the $\mathcal{L}_{csm}$ loss was added to the $\mathcal{L}_{ml}$ loss, while when we only used the $\mathcal{L}_{nsc}$ loss, the quality of the CAMs did not improve, but decreased by 6.4%. Only when $\mathcal{L}_{csm}$ and $\mathcal{L}_{nsc}$ were employed simultaneously, the quality improvement of the CAMs was the greatest among these three loss combinations, which improved the mIoU from 23.9% to 34.1%. We conjecture that, without the guidance of the common semantic features, it is hard for the network to converge during training by relying only on the non-common semantic information. This is because the non-common semantic information is not conducive to optimizing the classifier; on the contrary, it can interfere with classifier discrimination.

**Table 8.** The ablation research of different combinations of loss functions. We added $\mathcal{L}_{csm}$, $\mathcal{L}_{nsc}$, and $\mathcal{L}_{sl}$ in turn to observe the quality of the CAMs on the iSAID dataset by combining different loss functions. The mIoU (%) is the evaluation metric for CAM quality.

| $\mathcal{L}_{ml}$ | $\mathcal{L}_{sl}$ | $\mathcal{L}_{csm}$ | $\mathcal{L}_{nsc}$ | mIoU (%) |
|:---:|:---:|:---:|:---:|:---:|
| ✓ | | | | 23.9 |
| ✓ | ✓ | | | 26.7 |
| ✓ | ✓ | | ✓ | 20.3 |
| ✓ | ✓ | ✓ | | 29.8 |
| ✓ | ✓ | ✓ | ✓ | **34.1** |

## 5. Conclusions

This paper proposed a Cross-Image Semantic Mining (CISM) WSSS framework to discover more object regions and complete semantics in multi-category RS scenes with two novel loss functions: the CSM loss and the NSC loss. In particular, we extracted the prototype vectors from the class-specific features and further devised the PIE module to construct semantic similarity and difference. In addition, we proposed integrating the SLSC task and the corresponding novel loss into our framework to force the network to acquire knowledge from additional object regions. Furthermore, the MCSG module was designed to maintain a balanced distribution of samples across various categories and drastically increase the diversity of images. It was confirmed through extensive experiments that our method is able to resolve certain ambiguities and prevent false predictions by connecting correlated regions across images. Furthermore, our method generated more consistent and integral activation regions in the CAMs and provided new state-of-the-art results on the challenging iSAID dataset.

**Author Contributions:** Conceptualization, R.Z. and W.Z.; methodology, R.Z.; software, Z.Y.; validation, X.R., Z.Y. and R.Z.; formal analysis, R.Z.; investigation, X.R.; resources, W.Z., X.S. and K.F.; data curation, W.M. and R.Z.; writing—original draft preparation, R.Z.; writing—review and editing, Z.Y., X.R. and W.Z.; visualization, W.M. and R.Z.; supervision, W.Z.; project administration, W.Z.; funding acquisition, X.S. and K.F. All authors have read and agreed to the published version of the manuscript.

**Funding:** This research was funded by the National Science Fund for Distinguished Young Scholars under Grant 61725105 and the Surface of the State Natural Science Fund projects 62171436.

**Acknowledgments:** The authors would like to thank all their colleagues in the lab and the anonymous Reviewers for their very competent comments and helpful suggestions.

**Conflicts of Interest:** The authors declare no conflict of interest.

## Abbreviations

The following abbreviations are used in this manuscript:

| | |
|---|---|
| WSSS | Weakly Supervised Semantic Segmentation |
| FSSS | Fully Supervised Semantic Segmentation |
| SLSC | Single-Label Secondary Classification |
| CISM | Cross-Image Semantic Mining |
| CAM | Class Activation Maps |
| CSM | Common Semantic Mining |
| NSC | Non-common Semantic Contrastive |
| PIE | Prototype Interactive Enhancement |
| MCSG | Multi-Category Sample Generation |
| RS | Remote Sensing |
| NS | Natural Scene |

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
