# Peer review of "Weakly Supervised Semantic Segmentation in Aerial Imagery via Cross-Image Semantic Mining"

_remotesensing, doi:10.3390/rs15040986_

Round 1

Reviewer 1 Report

This paper proposes a cross-image weakly supervised semantic segmentation framework, uses image-level labels to mine the information of the common category of ground objects between different images, and generates CAM with higher reliability to obtain more accurate pseudo-labels. In general, the motivation of this article is novel, but there are some problems in writing and experiment. Specific comments are as follows:

1. The review of this paper is not sufficient. In the section of Weakly Supervised Semantic Segmentation, cross-image methods should be introduced, such as CIAN, RPNet et al. There is also a lack of an ECCV2020 paper “Mining Cross-Image Semantics for Weakly Supervised Semantic Segmentation”.

2. The meaning of some statements in the paper is unclear, for example, "Unlike the RPNet, we obtain the prototype vectors from the class-specific feature generated by the fully connected layer in classifier," at line 181 is inconsistent with the logic in the following text and Figure 1.

3. Figure 1 is not rigorous enough. In addition to generating the initial prototype vector, the class-agnostic feature is also used in the Masked Average Pooling (MAP) block of the refined prototype vector, not in the PIE module. The loss function and SLSC method need to be shown in the figure.

4. In the section of Prototype Interaction Enhancement, there is no formula to explain the operation after calculating the common category matching map A(i,j).

5. The comparison with other methods of supervision on the iSAID dataset seems unnecessary.

6. Only the visualization results of CIAN are available, and the visualization results of other representative comparison methods (such as RPNet) should be provided.

7. In Table 3, when using ResNet38 as the backbone, the model we proposed is not as good as the comparison method A2GNN. Is there any reason?

8. In table 8, it is necessary to add a loss function combination of ml + sl, and then the combination of ml + sl + csm and ml + sl + nsc, rather than the current combination.

9. There are some data errors. As shown in Table 7, Table 1 compares the indicators of baseline (CIAN) and CISM+MCSG, which is unreasonable, and the single CISM result are not matched with Table 5 and 6.

10. There are some text and format errors. For example, the name of the module in Figure 5 should be MSG instead of MSE; The meaning of F in formula 8 is incorrect; The bold items in Table 5 are incorrect; Line 450 describes MSG as MCSG; English capitalization (for example, are line165 Siamese, line244 B and b the same? TABLE is all capitalized, the uppercase and lowercase of section names are not unified)

Round 2

Reviewer 1 Report

In Fig. 6, "MSCG" should be "MCSG",please check and correct it.